# Kidney Measurement and Glomerular Filtration Rate Evolution in Children with Polycystic Kidney Disease

**DOI:** 10.3390/children11050575

**Published:** 2024-05-10

**Authors:** Ramona Stroescu, Mihai Gafencu, Ruxandra Maria Steflea, Flavia Chisavu

**Affiliations:** 1Department XI of Pediatrics—1st Pediatric Discipline, Center for Research on Growth and Developmental Disorders in Children, “Victor Babes” University of Medicine and Pharmacy Timisoara, Eftimie Murgu Sq. No. 2, 300041 Timisoara, Romania; 24th Pediatric Clinic, “Louis Turcanu” Children’s Clinical and Emergency Hospital, Iosif Nemoianu 2, 300011 Timisoara, Romania; steflea.ruxandra@umft.ro (R.M.S.);; 3Department XI of Pediatrics—3rd Pediatric Discipline, “Victor Babes” University of Medicine and Pharmacy Timisoara, Eftimie Murgu Sq. No. 2, 300041 Timisoara, Romania; 4Centre for Molecular Research in Nephrology and Vascular Disease, Faculty of Medicine “Victor Babes”, “Victor Babes” University of Medicine and Pharmacy Timisoara, 300041 Timisoara, Romania

**Keywords:** ADPKD, children, hyperfiltration, Schwartz, quadratic formula

## Abstract

Autosomal dominant polycystic kidney disease (ADPKD) is an inherited disorder characterized by renal tubular cystic dilatations. The cysts can develop anywhere along the nephron, and over time the cystic dilatation leads to kidney enlargement. On the other hand, the cysts begin to reduce the number of functional nephrons as a consequence of cystic expansion that further contributes to the decline in renal function over the years. The pressure exerted by the dilated cysts leads to compensatory mechanisms that further contribute to the decline in renal function. These structural changes are responsible of glomerular hyperfiltration states, albuminuria, proteinuria, and hematuria. However, the presentation of ADPKD varies in children, from a completely asymptomatic child with incidental ultrasound detection of cysts to a rapidly progressive disease. There have been reports of early onset ADPKD in children younger than 2 years that showed a more rapid decline in renal function. ADPKD is caused by a mutation in PKD1 and PKD2 genes. Today, the PKD1 gene mutation seems to account for up to 85% of the cases worldwide, and it is associated with worse renal outcomes. Individuals with PKD2 gene mutation seem to present a milder form of the disease, with a more delayed onset of end-stage kidney disease. The cardinal sign of ADPKD is the presence of renal cysts during renal ultrasound. The current guidelines provide clinicians the recommendations for genetic testing in children with a positive family history. Given that the vast majority of children with ADPKD present with normal or supra-normal kidney function, we explored the glomerular filtration rates dynamics and the renal ultrasound-adjusted percentiles. In total, 14 out of 16 patients had kidney percentiles over 90%. The gene mutations were equally distributed among our cohort. In addition, we compared the modified Schwartz formula to the quadratic equation after adjusting the serum creatinine measurements. It seems that even though children with ADPKD have enlarged kidneys, the renal function is more likely normal or near normal when the quadratic estimation of glomerular filtration rate is used (qGFR tended to be lower, 111.95 ± 12.43 mL/min/1.73 m^2^ when compared to Schwartz eGFR 126.28 ± 33.07 mL/min/1.73 m^2^, *p* = 0.14). Also, when the quadratic equation was employed, not even a single patient reached the glomerular hyperfiltration threshold. The quadratic formula showed that glomerular filtration rates are linear or slightly decreasing after 1 year of follow-up (quadratic ΔeGFR = −0.32 ± 5.78 mL/min/1.73 m^2^), as opposed to the Schwartz formula that can falsely classify children in a hyperfiltration state (ΔeGFR = 7.51 ± 19.46 mL/min/1.73 m^2^), *p* = 0.019.

## 1. Introduction

Autosomal polycystic kidney disease (ADPKD) is an inherited renal tubular disorder affecting 1 in 500–2500 people worldwide [1,2,3,4]. Although cyst development occurs early in life, even in utero in some cases [5,6], progressive renal function decline seems to be linked rather to the early hyperfiltration state of the kidney than the loss of kidney function [7]. However, glomerular hyperfiltration is associated with kidney enlargement, which in turn is a sign of progressive decline in renal function [8].

Kidney enlargement is the cardinal sign of progressive cystic dilatation of the renal tubules with variable echogenicity. This structural change in the renal parenchyma leads to a wide spectrum of complications: arterial hypertension [7], proteinuria [9], cerebral aneurysms [10], nephrolithiasis [11], hematuria [12], and urinary tract infections [13]. Glomerular filtration rate (GFR) decline occurs later in life, in the fourth or fifth decade of life [14], as a consequence of prolonged renal compensation [15].

Renal abnormalities in young adults with ADPKD are present even in near-normal GFR with modestly enlarged kidneys, including a decreased effective renal plasma flow and increased filtration flow and higher urinary albumin excretion [16]. However, prolonged hyperfiltration in children with ADPKD is associated with a faster decline in renal function [7].

The cardinal sign of ADPKD is the incidental finding of renal cysts detected by ultrasound scans regardless of the child’s age. Nevertheless, genetic testing should be performed in patients at risk for ADPKD when a family history of the disease is present and in patients with positive ultrasound testing [17]. While the inheritance pattern is autosomal dominant, the penetrance is incomplete with varying phenotypes [18]. The mutations occur in two genes, PKD1 (chromosome 16p13.3) and PKD2 (chromosome 4p21), which encode polycystin-1 (PC1) and polycystin-2 (PC2), respectively [19]. PKD1 mutations account for up to 85% of the cases worldwide, as compared to the 15–23.8% prevalence of PKD2 mutations [20,21].

Given the paucity of data regarding ADPKD epidemiology in children and associated renal alterations from childhood, we performed an observational retrospective study of children with ADPKD from west Romania. The main objective was to determine the rate of GFR decline between different age spectrums. The second objective was represented by the utility of personalized ultrasound kidney percentiles in assessing renal filtration rates in children with ADPKD. Also, we determined the degree of renal complications based on PKD gene mutations.

## 2. Material and Methods

### 2.1. Study Design

We conducted an observational retrospective cohort study of all patients admitted in a tertiary children’s hospital in west Romania. Data were retrieved from the Hospital’s Electronic System from 2014 until 2024.

Inclusion criterion was the presence of unilateral or bilateral renal cysts in the renal ultrasound evaluation with at least one serum creatinine measurement during admission. Exclusion criteria were all patients with renal cysts with etiology other than ADPKD, and patients with autosomal recessive PKD.

All patients were screened for renal cysts using 2D renal ultrasound performed by the same operator. Cyst size and number were recorded in each patient. All biological parameters were obtained during the first hospitalization and at follow-up visits in the same day the ultrasound was performed. Genetic testing and family history were noted during the first hospital admission. Follow-up was performed yearly using renal ultrasound, biological parameters, and anthropometric measurements. The final cohort consisted of 16 patients with ADPKD, with 8 patients followed-up for at least one year.

The study was approved by the Ethics Committee of the “Louis Turcanu” Emergency Hospital for Children, in Timisoara, Romania. Due to the retrospective nature of the study, informed consent was waived.

### 2.2. Outcomes and Definitions

The primary outcome was to determine the rate of estimated GFR (eGFR) decline in children with ADPKD. The serum creatinine was measured using the Jaffe method. In order to determine the compensated Jaffe method corrected to total serum protein concentration, we employed the following equation for all serum creatinine measurements, regardless of age [22]:Compensated Jaffe method=Serum creatinine (µmol/L)+24.91−0.3953×total protein concentration (g/L)

The compensated Jaffe serum creatinine levels were employed in the quadratic formula to estimate the GFR (qGFR) in children from 1 to 16 years old as follows [23]:

qGFR = 0.68 × (height/serum creatinine) − 0.0008 × (height/serum creatinine)^2^ + 0.48 × age − 21.53, in females

And,

qGFR = 0.68 × (height/serum creatinine) − 0.0008 × (height/serum creatinine)^2^ + 0.48 × age − 25.68, in males.

Height was measured in cm, serum creatinine in mg/dL, and age in years. For comparison reasons, we also employed the modified Schwartz formula in children aged 1 to 16 years old [24]:eGFR=0.413×height (cm)/serumcreatinine (mg/dL)

However, the modified Schwartz formula was developed for the enzymatic determination of serum creatinine, thus we applied the following equation to the corrected serum creatinine levels in order to determine the equivalent serum creatinine from the enzymatic method [22]:Serum creatinine by enzymatic method (µmol/L)=(Compensated Jaffe serum creatinine+0.1755)/0.9758

In children under 1 year old, the formula developed by Brion [25] was used to determine the eGFR as follows:eGFR=0.45×length (cm)/serum creatinine (mg/dL)

For conversion from µmol/l to mg/dl, a correction factor of 88.4 was performed.

Hyperfiltration was considered in children with higher eGFR than the ones reported in Table 1 [26,27,28].

The ultrasound was used to determine kidney length and parenchymal index, size, and number of the renal cysts. The kidney percentiles were calculated for age, height, and body surface area [29,30].

Proteinuria, quantified by 24 h urine collection, albuminuria (from spot urine), and hematuria were assessed as markers of kidney damage in children with ADPKD.

### 2.3. Variables of Interest

We noted gender, age, height, weight, family history, and serum creatinine measurements in the same day the kidney ultrasound was performed. Yearly follow-up was obtained in half the patients. The eGFRs were compared to kidney size, cysts, and affected genes.

### 2.4. Statistical Analysis

Data are expressed as means and standard deviation or medians and interquartile range for continuous variables, and numbers and percentages for categorical ones. The continuous variables were tested for normal distribution with the Shapiro–Wilk test. The statistical tests used for comparison between continuous variables with normal distribution were an independent *t*-test or an ANOVA, and for the ones without normal distribution, the Mann–Whitney or Kruskal–Wallis tests were appropriate. Categorical variables were evaluated with the Chi-square test. The correlation between specific continuous variables was evaluated using the Pearson’s correlation coefficient. A *p* value less than 0.05 was considered statistically significant. The analysis was performed using MedCalc^®^ Statistical Software version 22.021 (MedCalc Software Ltd., Ostend, Belgium; https://www.medcalc.org; Accessed on 1 March 2024).

## 3. Results

The cohort consisted of 16 children, with a predominance of the female gender (75%), with a mean age of 9.09 years with standard deviation of 5.74. Patient’s characteristics are noted in Table 2. All children were screened by renal ultrasound followed by genetic testing in 75%. PKD1 and PKD2 mutation were found in 37.5% of the cases. In total, 25% were not tested for the genetic mutation, yet half of them presented with a positive family history. Overall, 56.2% of the cohort had a positive family history of ADPKD. The youngest two patients (1 month and 6 months, respectively) had renal cysts detected intrauterine that underwent genetic testing in the first 6 months after birth with the PKD1 gene identified. After excluding the aforementioned infants, the average child’s age at the time of the diagnosis was 10.35 ± 4.93 years. However, the sub-analysis of patients with identified mutation showed that patients with PKD1 gene mutation were younger (*t*-test, PKD1 mean age = 5.59 ± 4.71 versus PKD2 mean age = 13 ± 5.13, *p* = 0.0265).

Renal ultrasound detected 56.25% of the children with bilateral renal cysts with a maximum cyst diameter of 0.75 cm (IQR 0.5–1.6 cm). The renal parenchyma index was 0.75 cm (IQR 0.7–0.83), with statistical differences among children with PKD1, PKD2, and who were untested. The average kidney length was 10.27 cm (IQR 9.2–11.27), with no differences between kidney length. However, kidney length differed when compared to the gene mutation: children with PKD1 gene mutation had a mean kidney length of 9.57 cm (IQR 6.88–10.25), much lower when compared to those with PKD2 gene mutation, who had a mean length of 11.27 cm (IQR 10.95–11.62).

We analyzed kidney size using percentiles for age, height, and body surface area, as seen in Table 2. There were no statistical differences between the three percentiles. The mean percentile in children with ADPKD from our cohort was over 90%, regardless of the percentile used.

The mean serum creatinine level at diagnosis was 40.06 ± 17.54 µmol/L. After performing the correction of the serum creatinine to protein concentration, the mean compensated serum creatinine was 39.67 ± 17.54 µmol/L. The enzymatic serum creatinine level of 40.83 ± 17.98 µmol/L was estimated using the compensated serum creatinine level after applying the correction for total serum proteins. There were no differences between the methods used to measure the serum creatinine levels, as all children had normal serum protein levels. However, even though there were no statistical differences between eGFR and qGFR, qGFR tended to be lower, 111.95 ± 12.43 mL/min/1.73 m^2^ compared to Schwartz eGFR 126.28 ± 33.07 mL/min/1.73 m^2^, *p* = 0.14. In addition, the GFR values were similar in both identified genetic mutations, even in those whom were not tested (Table 3).

Based on estimations in GFR according to the Brion, Schwartz, and quadratic formulas, the rates of glomerular hyperfiltration (GHF) were noted in both the initial presentation and in the 1-year follow-up (Appendix A). On the one hand, infants under 1 year old did not reached the hyperfiltration state at the initial presentation; however, at the 1-year follow-up, the Schwartz formula classified both children with GHF, as opposed to the quadratic formula, where none reached the hyperfiltration thresholds. On the other hand, four out of the six children over 2 years were classified as having GHF rates (over 135 mL/min/1.73 m^2^), as opposed to the quadratic formula, where none of the children reached the GHF threshold. Interestingly, at the 1-year follow-up, there were still four out of six children with GHF when the Schwartz formula was applied and still no patient in the GHF state when quadratic formula was employed.

The most common complications were represented by urinary tract infections (50%), followed by nephrolithiasis and vesical-ureteral reflux in 12.5% of the cohort. Markers of kidney damage were also present at the time of the diagnosis, with microscopic hematuria being present in 31.25% of the children followed by proteinuria in 12.5% of the cases.

Another aspect that was taken under consideration was the presence of other genetic anomalies in children with ADPKD. Three patients with PKD2 gene mutation had another genetic disorder that overlapped ADPKD. Two sisters had Charcot–Marie–Tooth neuropathy, and one girl had a Wilms tumor and underwent total right nephrectomy 3 years prior to the diagnosis of ADPKD.

Follow-up was performed in half of the patients. Unexpectedly, when we employed both GFR equations, the Schwartz formula tended to overestimate the GFR, with a marked increase in the estimation when compared to the quadratic formula. These results underline that even though kidney enlargement was present, the renal function seemed to be preserved when the quadratic formula was applied (Figure 1). These results were further analyzed as delta differences between the estimated GFRs. At the initial presentation and at follow-up, the mean GFR was overestimating the renal function when comparing the Schwartz formula to the quadratic one (137.55 versus 118.08 mL/min/1.73 m^2^ at initial presentation and 145.05 versus 117.75 mL/min/1.73 m^2^ at the 1-year follow-up). While the Schwartz formula proved to increase over 1 year with a ΔeGFR = 7.51 ± 19.46 mL/min/1.73 m^2^, the quadratic formula showed an annual decrease in eGFR of 0.32 ± 5.78 mL/min/1.73 m^2^ (*p* = 0.019).

## 4. Discussion

With this study, we estimated the rates of glomerular filtration in children with ADPKD using adapted formulas for age, Brion, and Schwartz equations, respectively. In addition, we employed the quadratic formula in children over 1 year old for comparison reasons. Almost all children had enlarged kidneys at the time of the diagnosis. However, kidney size did not correlate with hyperfiltration states. Although there were no statistical differences between Schwartz and quadratic eGFR values, the Schwartz formula seems to falsely classify patients in a GHF state when compared to the quadratic equation. Markers of kidney damage and complications occur early in childhood.

The age distribution was heterogeneous, from 1 month until 16 years old, with a predominance of the female gender similar to published data [18]. The very early onset of ADPKD was similar to the literature reports (12%) [31]. The incidental finding of cysts during renal ultrasound, kidney size, and positive family history led to the genetic testing of these patients. Due to the small size of the cohort, the identified mutations were equally distributed, even though the literature reports PKD1 gene mutations in over 85% of the ADPKD patients [20,21].

Overall, kidney enlargement was estimated using kidney length percentiles adjusted for age, height, and body surface area. Even though over 80% of the children had, at the time of the diagnosis, increased kidney size, evidenced by 90% percentile, PKD1 mutation seemed to associate lower kidney size and kidney length percentiles when compared to PKD2 mutations, but without reaching statistical significance. This is important, as these patients also had lower age, smaller renal cysts, and reduced parenchymal index. However, these results should be interpreted cautiously, as children with PKD1 were indeed younger and thus had lower kidney size compared to PKD2 ones, yet had higher when kidney percentiles were employed. It seems that PKD1 gene mutation is associated with worse renal outcomes [20]. As it was previously showed, children with PKD1 mutation are diagnosed as early as in utero [19,32], as it was the case in 2 out of 6 patients with PKD1 mutation from our cohort.

Although this is a small-sized study, the family history was positive in over 50% of the cases with a 75% genetic testing rate. Currently, the ADPKD guidelines in children under 18 years old recommend genetic testing to be deferred in asymptomatic at-risk children without renal cysts on ultrasound [17]. However, renal ultrasound cannot exclude ADPKD [33], and it can only be used as a complementary tool in children at risk of or with ADPKD. Albeit family history proved to have a prognostic value in patients with ADPKD [20], in families with a known genotype, genetic testing is most likely informative [21]. However, in our cohort, all children had incidental renal cysts detected during ultrasound doubled by increased renal size when renal percentiles were calculated.

The incidental finding of ADPKD was linked to the recurrent UTI episodes that were investigated by renal ultrasound. Also, the three patients with another genetic disease were diagnosed after the incidental finding of the renal cysts on ultrasound. In addition, the presence of the PKD2 gene mutation in these patients could be an indicator of a milder disease course [20].

The utility of serum creatinine in stable renal function has made this endogenous marker to be used worldwide as a marker of kidney function. The revised Schwartz formula is the most commonly used equation in children over 1 year old [24]. However, previously published reports showed that in children with a GFR over 90 mL/min/1.73 m^2^, the Schwartz formula loses its accuracy [23,34,35]. In addition, creatinine-based formulas for estimating GFR are dependent of the accuracy of the serum creatinine measurement. The currently used Schwartz formula is applicable when serum creatinine is determined using the enzymatic method [24]. This is why we used the corrected compensated Jaffe serum creatinine to estimate the enzymatic serum creatinine that we used in GFR estimations in all children. Furthermore, we employed the quadratic eGFR in children over 1 year old, as this formula has been validated against inulin clearance in children with renal failure and also in children with normal or supra-normal GFR (including hyperfiltration) [23].

Our results are contradictory, even though they failed to reach statistical significance. It seems that Schwartz formula overestimates the eGFR when compared to the quadratic formula. Even more, the quadratic formula proved that the loss of kidney function is evident after the 1-year follow-up as opposed to the Schwartz formula that tends to continue in a GHF state, reaching statistical significance even in a reduced-size cohort like ours. Although the GHF thresholds derive from systematic reviews, the most recent review by Pottel from 2022 gathered the GHF threshold in children [36]. The consensus GHF rates are categorized by age, reducing the bias in children under 2 years old. With this small-sized study, we draw attention on the false GHF state induced by the Schwartz formula. The serum creatinine levels did not increase over time, yet the children had a linear growth spur leading to the overestimation of GFR when Schwartz formula was employed. The quadratic equation takes into account the age and the sex of the patients besides the height and serum creatinine level from the Schwartz formula [23,24].

Kidney enlargement was classically determined by the total kidney volume in both adults and children [4,15,16]. The use of kidney percentiles combined with genotype can be a predictor for rapid progression in ADPKD, as previously published by Chen [37]. In our study, even though all children presented with kidney enlargement, the GHF thresholds were reached only in children over 1 year old when the Schwartz formula was used. Although renal length was over the 90% percentile, there was no correlation between renal length and eGFR, as was previously shown [8]. Interestingly, when the Schwartz formula was used to estimate GFR, GHF was more frequently seen in patients with PKD1 gene mutation rather than PKD2, consistent in the 1-year follow-up. However, these results should be interpreted cautiously, mostly due to the small size and also the overestimation of Schwartz eGFR when compared to qGFR, as we proved that Schwartz can falsely classify children with ADPKD in the GHF state.

The main limitation of our study is related to the small cohort size and the reduced follow-up. To our knowledge, this is the first study from an Eastern European country that evaluates the GFR dynamics in children with ADPKD. In addition, this is the first study that showed that quadratic GFR could be used as a tool for a more precise renal function evaluation.

In conclusion, the eGFR is more likely to be normal or near normal in children under 18 years old with ADPKD when quadratic GFR is used. Although children with ADPKD have increased renal length, this is not associated with a hyperfiltration rate. In addition, quadratic eGFR showed that the glomerular filtration rate is linear or even decreasing, as opposed to Schwartz, that is most likely prone to overestimate eGFR. Future studies with larger cohorts are needed to validate the use of quadratic GFR formula in children at risk for GHF.

## Figures and Tables

**Figure 1 children-11-00575-f001:**
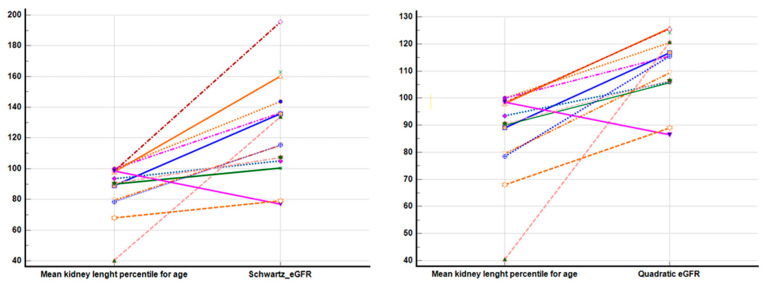
Mean kidney length percentiles adjusted for age and renal function Legend: Mean kidney length percentile for age was assessed using renal ultrasound length according to the children’s age. Different colored lines represent the correspondence between kidney size and glomerular filtration rates using the Schwartz and quadratic formulas.

**Table 1 children-11-00575-t001:** Glomerular hyperfiltration according to age.

Age (years)	Glomerular Hyperfiltration Thresholds
≤0.10	69.6
0.10–0.30	89.7
0.30–0.66	98.9
0.66–1.00	116.5
1.00–1.50	126.4
1.50–2.00	130
>2.00	135

Glomerular hyperfiltration is measured in mL/min/1.73 m^2^. The hyperfiltration GFRs thresholds were calculated as mean + 1.96 × standard deviation in children under 2 years old. Over the age of 2 years old, it is considered the above value of the GFR to reflect the glomerular hyperfiltration rates.

**Table 2 children-11-00575-t002:** Patient’s baseline characteristics at first presentation.

Parameter	PKD1 N = 6	PKD2 N = 6	NA N = 4	Total N = 16	*p* Value
Sex—female	4 (66.7%)	5 (83.3%)	3 (75%)	12 (75%)	0.711 ^1^
Age years A + SD	5.59 (4.71)	13 (5.13)	8.5 (5.32)	9.09 (5.74)	0.07 ^2^
Intrauterine renal cysts	2 (33.3%)	-	-	2 (12.5%)	0.808 ^1^
Age at diagnosis A + SD	5.59 (4.71)	13 (5.13)	8.5 (5.32)	9.09 (5.74)	0.07 ^2^
Family history	4 (66.7%)	4 (66.7%)	1 (25%)	9 (56.2%)	0.226 ^1^
Patients with cysts on both kidneys	4 (66.7%)	3 (50%)	2 (50%)	9 (56.2%)	0.803 ^1^
Maximal cyst diameter cm M + IQR	0.65 (0.3–0.8)	1.3 (0.4–2)	1.1 (0.8–1.6)	0.75 (0.5–1.6)	0.306 ^3^
Parenchymal index cm, M + IQR	0.7 (0.45–0.72)	0.725 (0.7–0.8)	0.9 (0.87–0.95)	0.75 (0.7–0.83)	0.01 ^3^
Average kidneys length cm M + IQR	9.57 (6.88–10.25)	11.27 (10.95–11.62)	10.17 (9.77–11.18)	10.27 (9.2–11.27)	0.14 ^3^
Percentiles per age M + IQR	88.75 (57–100)	90 (75.87–92.5)	96 (91.25–99.25)	90.5 (78.75–98.5)	0.503 ^3^
Percentiles per height M + IQR	86.75 (44.5–100)	91 (83–98.87)	97 (93–100)	94 (80–100)	0.558 ^3^
Percentiles per BSA M + IQR	77.5 (49–100)	96.5 (81.87–98.5)	96.5 (77.5–98.75)	96.5 (61.62–99.87)	0.892 ^3^

Legend: N = number expressed as value and percentage; A = average; SD = standard deviation; M = median; IQR = interquartile range; cm = centimeters; BSA = body surface area. ^1^ Chi-square test; ^2^ ANOVA; ^3^ Kruskal–Wallis test.

**Table 3 children-11-00575-t003:** The estimated glomerular filtration rates after adjusting the serum creatinine.

Parameter	PKD1 N = 6	PKD2 N = 6	NA N = 4	Total N = 16	*p* Value
Serum creatinine at diagnostic Jaffe unadjusted μmol/L A + SD	30.5 (11.82)	46.66 (18.44)	44.5 (21.04)	40.06 (17.54)	0.249 ^1^
Serum creatinine at diagnostic Jaffe adjusted to proteins μmol/L A + SD	38.54 (9.37)	46.27 (18.44)	44.11 (21.04)	42.83 (15.61)	0.71 ^1^
Serum creatinine at diagnostic enzymatic equivalent μmol/L A + SD	39.67 (9.61)	47.6 (18.9)	45.38 (21.56)	44.07 (16)	0.71 ^1^
eGFR mL/min/1.73 sm A + SD (N = 14—patients older than 2 years) (Quadratic)	117.81 (7.07)	111.06 (13.68)	107.42 (15.24)	111.95 (12.43)	0.521 ^1^
eGFR mL/min/1.73 sm A + SD (N = 14—patients older than 2 years) (Schwartz)	136.42 (18.48)	126.76 (43.59)	115.42 (30.59)	126.28 (33.07)	0.702 ^1^

Legend: μmol/L = micromoles per liter; eGFR = estimated glomerular filtration rate; NA = not assigned; N = number expressed as value and standard deviation (SD); ^1^ ANOVA.

## Data Availability

The raw data supporting the conclusions of this article will be made available by the authors on request.

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
