# Peer review of "Kidney Measurement and Glomerular Filtration Rate Evolution in Children with Polycystic Kidney Disease"

_children, 2024, doi:10.3390/children11050575_

Round 1

Reviewer 1 Report

Comments and Suggestions for Authors

Thank you for the opportunity to review the manuscript. The authors conducted a retrospective study on ADPKD to assess the clinical condition in terms of GFR, ultrasound kidney findings and renal complications. There are several comments / concerns.

1.      Line 132: Is the kidney length percentile based on the average size of both kidneys? Or just the kidney with cysts in patients with unilateral pathology?

2.      Line 163 Table 2: Intrauterine renal cysts in patients with PKD1 or PKD2 patients? (different in text and table)

3.      Line 204: 12.5% of patients had VUR. Are these investigations done for all patients? Or just for those for UTI?

4.      Line 235: Authors mentioned that Schwartz formula over-estimated GFR and falsely classify patients in hyperfiltration state, assuming the quadratic formula is the ‘gold standard’ reflecting the real GFR. In Gao et al. original paper (PMID 23254901), it is mentioned that this quadratic formula is accurate for children with moderate renal failure, *but not for those with less renal impairment or hyperfiltration. The mean eGFR of this cohort >110, thus the result can just show the difference between the two methods of eGFR, but not to conclude which is more accurate / overestimate hyperfiltration state. Indeed, estimation of eGFR using formula in patients in ADPKD is often inaccurate (Rodriguez et al. PMID 35357684). Thus I cannot agree with the conclusion that this study proved the importance of quadratic GFR as a tool for more precise renal function evaluation (Line 306).

Comments on the Quality of English Language

No major issues

Some typo e.g. Line 115 Creatinine

Author Response

  1. Line 132: Is the kidney length percentile based on the average size of both kidneys? Or just the kidney with cysts in patients with unilateral pathology?

A: The kidney length percentile was based on the average size of both kidneys. The ultrasound is more accurate for kidney size in children with ADPKD rather than assessing the presence of small kidney cysts (less than 4 mm).

  1. Line 163 Table 2: Intrauterine renal cysts in patients with PKD1 or PKD2 patients? (different in text and table)

A: It was a typo. We modified the text accordingly. Intrauterine renal cysts were observed in 2 patients with PKD1.

  1. Line 204: 12.5% of patients had VUR. Are these investigations done for all patients? Or just for those for UTI?

A: The diagnosis of VUR using retrograde cystography was performed only in 2 patients that besides the recurrent UTI, also had a degree of hydronephrosis. This is a retrospective study and perhaps, in children, the incidence of VUR is higher among ADPKD patients.

  1. Line 235: Authors mentioned that Schwartz formula over-estimated GFR and falsely classify patients in hyperfiltration state, assuming the quadratic formula is the ‘gold standard’ reflecting the real GFR. In Gao et al. original paper (PMID 23254901), it is mentioned that this quadratic formula is accurate for children with moderate renal failure, *but not for those with less renal impairment or hyperfiltration. The mean eGFR of this cohort >110, thus the result can just show the difference between the two methods of eGFR, but not to conclude which is more accurate / overestimate hyperfiltration state. Indeed, estimation of eGFR using formula in patients in ADPKD is often inaccurate (Rodriguez et al. PMID 35357684). Thus I cannot agree with the conclusion that this study proved the importance of quadratic GFR as a tool for more precise renal function evaluation (Line 306).

A: Thank you or your kind remark. Indeed, Gao reported quadratic formula to be a better glomerular filtration estimation compared to the revised Schwartz formula. She validated the formula using the inulin clearance (which is the gold standard for GFR measurement). In her study, she clearly specified that higher GFRs than 103ml/min/1.73sm measured with inulin, the quadratic formula was significantly better than the Schwartz formula (this is the cut of value of height/serum creatinine of 251). She also stated that quadratic estimation of GFR had better performance in estimating GFR compared to Schwartz at measured GFRs higher than 103ml/min/1.73sm (p=0.02 for 10% accuracy and p=0.001 for 20% accuracy). We are aware that Gao’s cohort was mixed and she did not evaluated strictly patients with ADPKD. On the other hand, ADPKD is a known status of hyperfiltration. With our results, we wanted to draw a line in the sand regarding the hyperfiltration status in children with ADPKD. Maybe the real moment of hyperfiltration appears later in life. Nevertheless, Rodriguez could not find a suitable GFR estimation in adults with ADPKD. However, even though our cohort is small, we found higher eGFR’s in the 1 year follow-up when Schwartz formula is used, thus we believe that quadratic formula is more suitable for patients whom are prone to hyperfiltration states. We modified the conclusions to a more suitable manner.

Reviewer 2 Report

Comments and Suggestions for Authors

The Authors explored the glomerular filtration rates dynamics and the renal ultrasound adjusted percentiles in 16 children with ADPKD. 14 out of 16 children (6 with PKD1 and 6 with PKD2 mutations) had kidney percentile over 90%. By comparing the modified Schwartz formula to the quadratic equation after adjusting the serum creatinine measurements, the Authors found that even though children with ADPKD have enlarged kidneys, the renal function is more likely normal or near normal when the quadratic estimation of glomerular filtration rate was used.

General comment:

This is an good paper which adds a new knowledge on PKD1 and PKD2 genetically determined ADPKD in infant age, and what we likely can do as methodology for improve the long-term prognosis of these patients. I have few comments

Minor comment

1) The Authors write in Discussion on lines 245-251 “Overall, kidney enlargement was estimated using kidney length percentile adjusted for age, height, and body surface area. Even though, over 80% of the children had at the time of the diagnosis increased kidney size, evidenced by 90% percentile, PKD1 mutation seemed to associate lower kidney size and kidney length percentiles when compared to PKD2 mutations, but without reaching statistical significance. This is important, as these patients also had lower age, smaller renal cysts and reduced parenchymal index. It seems that PKD1 gene mutation is associated with worse renal outcomes [21].”

Based on kidney length percentiles it seems that the lower kidney size in PKD1 (see Table 2, last three rows) reflected the enlargement due to ADPKD pathology in kidney with a reduced basal mass. In effect, patients with ADPKD tipe 1 have a worse prognosis. Can the Authors further comment on? Could they support this consideration with literature data?

Comments on the Quality of English Language

good, but it could be improved

Author Response

Thank you for your kind remark. We performed a sub analysis of patients mutation and age and it seems that indeed patients with PKD1 mutation are younger thus they have smaller kidneys when compared to PKD2 children, yet higher for age, height and BSA kidney size percentiles. Our results are in accordance with previously published data as they were diagnosed earlier in life, even from in utero. We modified the discussions using your suggestion and also added the aforementioned results in the results section. In addition, we completed the reference list.

Round 2

Reviewer 1 Report

Comments and Suggestions for Authors

The authors have addressed my main concerns.

Minor issues

1. Table 1: What is the hyperfiltration threshold for patient 1 - 1.5 years old

2. Minor typo including line 115 "creatinine"

Author Response

Thank you for your remarks. We completed the table regarding hyperfiltration thresholds and the typos. 

Reviewer 2 Report

Comments and Suggestions for Authors

I have no further comments

Comments on the Quality of English Language

good

Author Response

Thank you for your contribution for the improvement of the article.